# Position: Iterative Online-Offline Joint Optimization is Needed to Manage Complex LLM Copyright Risks

Yanzhou Pan [* 1]  Jiayi Chen [* 2]  Jiamin Chen [3]  Zhaozhuo Xu [4]  Denghui Zhang [4]

## Abstract

The infringement risks of LLMs have raised significant copyright concerns across different stages of the model lifecycle. While current methods often address these issues separately, this position paper argues that the LLM copyright challenges are inherently connected, and independent optimization of these solutions leads to theoretical bottlenecks. Building on this insight, we further argue that managing LLM copyright risks requires a systemic approach rather than fragmented solutions. In this paper, we analyze the limitations of existing methods in detail and introduce an iterative online-offline joint optimization framework to effectively manage complex LLM copyright risks. We demonstrate that this framework offers a scalable and practical solution to mitigate LLM infringement risks, and also outline new research directions that emerge from this perspective.

## 1. Introduction

The application of large language models (LLMs) has brought significant benefits and transformative changes across various domains (Hadi et al., 2023; Raiaan et al., 2024). However, this advancement also introduces complex copyright risks and ethical challenges (Laakso, 2023; Jiao et al., 2024; Liu et al., 2024b; Zhang et al., 2025). LLMs are typically trained on extensive datasets that often include copyrighted material, which can inadvertently lead to the reproduction of protected content, resulting in legal, ethical, and reputational concerns (Panaitescu-Liess et al., 2025). Recently, several high-profile lawsuits have been filed regarding the use of copyright-protected data for training models without permission. Notable cases include The New York Times vs. OpenAI/Microsoft (New York Times,

2023), Authors Guild vs. OpenAI (USAuthorsGuild, 2023), and Getty Images vs. Stability AI (Davies & Dennis, 2024). These legal cases highlight the critical need for effective measures to manage these copyright risks.

Preventing copyright infringement in LLMs is complex for several reasons. First, copyright laws vary by regions and evolve over time, making it difficult for LLMs to accurately determine the legal status of a given content. Second, user inputs differ in content and intent, affecting LLM behavior (Zhao et al., 2024; Mueller et al., 2024). Recent studies (Liu et al., 2024a; Xu et al., 2024) show that LLMs respond with varying compliance levels to different probing prompts and often fail to respect the copyrighted notices. Third, the vast training data complicates screening for copyrighted content, as many datasets lack clear metadata or authorization details, making this copyright validation process highly resource-intensive. Lastly, timely and accurate responses to potential infringements are essential to avoid legal and financial risks.

Researchers have made numerous attempts to tackle these copyright issues. Some of them focus on LLM online serving stages, such as system prompts (Xie et al., 2023; Xu et al., 2024), output filtering (Ziegler, 2021; Liu et al., 2024a) and decoding-time methods (Ippolito et al., 2023; Shi et al., 2024; Flemings et al., 2024). Others mitigate copyright issues during the LLM offline development stages, such as machine unlearning (Dou et al., 2024; Yao et al., 2024; Yu et al., 2023; Eldan & Russinovich, 2023; Chen & Yang, 2023), data cleansing (Ladhak et al., 2023), combating data poisoning (Huang et al., 2024; Yan et al., 2024), and training-based methods (Mireshghallah et al., 2023; Li et al., 2022; 2024; Chu et al., 2024). However, existing methods primarily address isolated aspects of copyright issues, overlooking the interconnected risks across different stages of the LLM lifecycle. For instance, copyright violations during the serving stage often stem from the model memorizing copyrighted data during the training stage (Wei et al., 2024a; Karamolegkou et al., 2023; Nasr et al., 2025; Vyas et al., 2023; Zhao et al., 2024). Based on these observations and the guidance from U.S. Government Copyright Office (U.S. Copyright Office, 2023; 2024), this position paper focuses on the copyright law and policy issues raised by LLM, including the scope of copyright in LLM-generated

---

*Equal contribution ¹Google LLC ²National University of Singapore ³Northeastern University ⁴Stevens Institute of Technology. Correspondence to: Denghui Zhang <dzhang42@stevens.edu>.

*Proceedings of the 42ⁿᵈ International Conference on Machine Learning*, Vancouver, Canada. PMLR 267, 2025. Copyright 2025 by the author(s).

works and the use of copyrighted materials in LLM training. **We argues that LLM copyright risks are interconnected and cannot be adequately addressed through isolated optimization efforts.** Instead, a holistic, jointly optimized framework is necessary to effectively manage these risks.

To support our position, we categorized existing methods into online and offline approaches and analyzed their strategies for mitigating copyright infringement. We highlighted the limitations of these methods, emphasizing that independent optimization cannot effectively address the interconnected nature of LLM copyright issues. To overcome these bottlenecks, we propose a comprehensive framework encompassing three components: mitigation, examination, and calibration. The mitigation component addresses immediate risks using online methods. The examination component bridges mitigation and calibration, analyzing issues from both the online and training phases. Finally, the calibration module acts as the centralized optimizer, using examination results to improve the entire system. These components work together iteratively, enabling continuous optimization and mutual enhancement.

This paper is structured as follows: Section 2 provides a comprehensive classification and summary of current works. Section 3 offers a detailed analysis of the limitations faced by current approaches and argues that isolated optimization of the infringement mitigation approaches cannot overcome these bottlenecks. Section 4 talks about several alternative perspectives and highlights areas that merit further discussion. Section 5 proposes a novel online-offline unified framework for managing complex LLM copyright risks. We explain the modules within this framework and their interactions, demonstrating its comprehensiveness and effectiveness in addressing LLM copyright issues. Finally, the conclusion is provided in Section 6.

## 2. Overview of Current Works

As shown in Figure 1, copyright risks associated with LLMs can emerge at various stages of their lifecycle (Lee et al., 2024; Kretschmer et al., 2024). In the online phase, these risks may arise from user inputs or the model's outputs. In the offline phase, risks may occur during data collection (Min et al., 2024) or processing. To address issues occurring at different stages, various existing approaches have been proposed. Based on the stage of the LLM lifecycle where these methods are applied, we classify them into two categories: online approaches and offline approaches.

**Online approaches.** Some approaches focus specifically on mitigating infringement in the output generated during the online serving phase of the model. We refer to these approaches as *online approaches*. Specifically, common online approaches can be divided into the following three

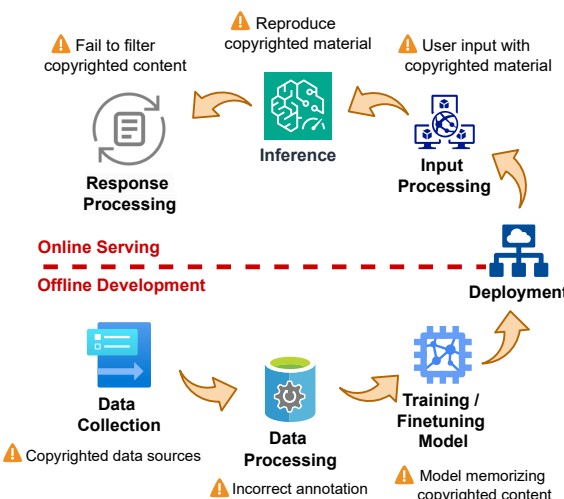

*Figure 1.* The LLM lifecycle can be divided into offline development stage and online serving stage. Copyright risks can arise at various parts throughout this lifecycle.

categories: *(i) System prompt* approaches focus on tailoring the system prompt to guide the model to generate responsible, non-harmful responses, and to prevent the generation of copyrighted content. For example, Xie et al. (Xie et al., 2023) introduced self-reminder prompts as an effective strategy to mitigate jailbreak attacks, which can enable LLMs to bypass ethical safeguards and generate harmful responses. *(ii) Output filtering* approaches, such as GitHub Copilot (Ziegler, 2021; GitHub, 2025) and SHIELD (Liu et al., 2024a), employ filtering and blocking mechanisms to detect and avoid generating outputs similar to copyright-protected materials. Specifically, GitHub Coplit provides "block suggestions matching public code" option for users to filter out the similar codes in the training set and SHIELD introduced an agent-based method to detect and verify the copyrighted content in the model's output, thereby preventing the generation of copyrighted material. *(iii) Decoding-time* approaches refer to techniques where copyright takedowns happen during the decoding phase, including MemFree (Ippolito et al., 2023) and R-CAD (Shi et al., 2024). MemFree prevented the verbatim regurgitation of blocklisted content by n-gram match and token selection. R-CAD reversed the Context-aware decoding(CAD) process to down-weight the blocklisted materials.

**Offline approaches.** In contrast, other approaches are employed during the model development and training phase, proactively addressing potential copyright concerns before the model is deployed. We categorized these approaches as *offline approaches*. The following are three major types of offline approaches: *(i) Machine unlearning* (Dou et al., 2024; Yao et al., 2024; Yu et al., 2023; Eldan & Russinovich,

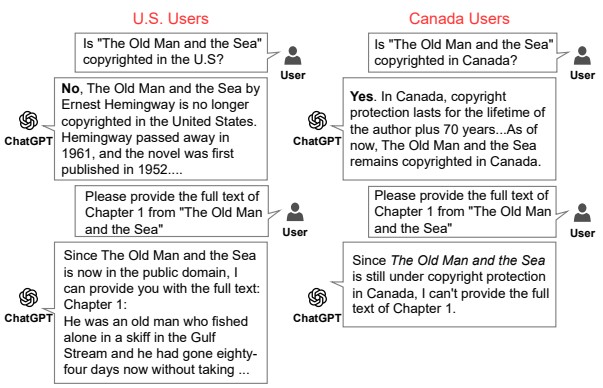

*Figure 2.* An example of ChatGPT incorrectly recognizing The Old Man and the Sea as being in the public domain in the U.S. and copyrighted in Canada, which is the exact opposite of the actual situation.

2023; Chen & Yang, 2023) methods enable models to forget the blocklisted materials encountered during training, reducing their occurrence in the output by using an unlearn set and retain set. Various metrics have been proposed to evaluate unlearning effectiveness. For instance, Kurmanji et al., (Kurmanji et al., 2023) introduce metrics targeting bias removal, confusion resolution, and privacy. *(ii) Data cleansing and combating data poisoning* approaches focus on ensuring the quality and integrity of training data. Data cleansing (Ladhak et al., 2023) involves tracing, identifying, and removing specific training instances that could result in undesirable outputs. Combating data poisoning (Mireshghallah et al., 2023; Li et al., 2022) aims to prevent malicious modifications to the training data such as backdoors, thereby safeguarding against potential copyright violations. *(iii) Training-based* (Mireshghallah et al., 2023; Li et al., 2022; 2024; Chu et al., 2024) approaches are introduced by modifying training procedures for adapting models to align with specific ethical, legal, or private constraints. These methods adapt the model's behavior during training to comply with predefined guidelines and restrictions.

## 3. The Pitfalls of Isolated Infringement Mitigation Approaches

Although many online and offline approaches have been designed to address copyright issues in LLMs, they overlook the interconnections between these risks. As a result, while some approaches succeed in mitigating specific risks, relying on isolated optimizations introduces limitations that hinder comprehensive copyright risk management. This section highlights the shortcomings of relying solely on either online or offline approaches.

*Table 1.* Evaluation results from CopyBench (Chen et al., 2024) compare LLMs using different online infringement mitigation strategies, such as System Prompts and MemFree Decoding. Positive percentage indicates that infringement has been mitigated, while negative percentage indicates increased severity.

| LMs | Literal (% ↓) | Events (% ↓) | Characters (% ↓) |
|---|---|---|---|
| Llama2-13B | 0.1 | 0.3 | 2.0 |
| +System Prompts | 0.0 (-50%) | 0.5 (+33%) | 2.0 (0%) |
| +MemFree Decoding | 0.0 (-100%) | 0.3 (0%) | 2.0 (0%) |
| Llama2-70B | 2.4 | 4.0 | 10.3 |
| +System Prompts | 2.6 (+7%) | 4.7 (+18%) | 11.5 (+11%) |
| +MemFree Decoding | 0.3 (-87%) | 3.8 (-4%) | 10.9 (+5%) |
| Llama2-70B-Tulu | 1.0 | 2.8 | 4.6 |
| +System Prompts | 0.7 (-26%) | 2.0 (-28%) | 3.3 (-29%) |
| +MemFree Decoding | 0.1 (-91%) | 2.9 (+2%) | 4.4 (-5%) |
| Llama3-70B | 10.5 | 6.9 | 15.6 |
| +System Prompts | 11.0 (+5%) | 5.9 (-14%) | 15.0 (-4%) |
| +MemFree Decoding | 0.6 (-94%) | 7.2 (+5%) | 15.5 (0%) |

### 3.1. Limitations of Online-Only Approaches

Online approaches are limited to intervening during the serving stage and cannot access or modify the training data or influence the training process. This constraint significantly reduces their ability to address issues originating from the offline model development stage. Due to this limitation, there are certain copyright risks that cannot be resolved.

**Localized copyright status.** Copyright regulations differ from country to country (Reindl, 1997). Certain contents may have different copyright statuses depending on the jurisdiction, complicating the task for LLMs in effectively managing copyright risks. For example, *The Old Man and the Sea* (Hemingway, 1952) remains copyrighted in the U.S. until 2047 (U.S.CopyrightOffice, 2022), but entered the public domain in Canada in 2011 (CanadianIntellectualProperty Office, 2024). Ideally, the LLM should avoid outputting the original content for U.S. users and allow it for Canada users. However, a simple experiment with ChatGPT showed the opposite. As shown in Figure 2, U.S. users (left side) received the copyrighted content after several attempts, posing potential infringement risks, while Canada users (right side) encountered an overly protective result, despite being entitled to access the content. A recent study (Liu et al., 2024a) further highlights this challenge by introducing a "Partially Copyrighted" dataset (BS-PC) to test online mitigation strategies across LLMs. Their results clearly indicate that existing online-only methods are insufficient to address the complexities of this issue. To address the gap, offline support like maintaining a dynamic and localized database is required.

**Non-literal copying.** Non-literal copying happens when a model rephrases or paraphrases content, changing the wording or structure while retaining the original meaning. CopyBench (Chen et al., 2024) evaluates both literal and non-literal reproduction of copyrighted content in LLMs

with various online-only mitigations, including MemFree (Ippolito et al., 2023) and System Prompts. Table 1 presents the outcomes of their evaluation of some online approaches. The results indicate that while MemFree can prevent literal copying to some extent, neither System Prompts nor the MemFree can effectively reduce non-literal copying (events and characters). Non-literal copying remains a significant concern because it still poses a threat to copyright, especially when the generated content may appear novel but still contains underlying copied ideas or structures. The key challenge in detecting non-literal infringement lies in the fact that such content often avoids direct duplication, instead using alternative vocabulary or structural variations to convey the same ideas. Due to the nature of non-literal copying, online approaches struggle to detect and mitigate it effectively, highlighting the need for additional offline support such as model fine-tuning and training data auditing.

**Fail to trace back.** Online approaches are unable to address issues stemming from the training process or training data, as they primarily operate during the online inference stage and do not intervene in the training phase. For instance, the use of copyrighted works in training data can lead to legal challenges if rights holders believe their work has been used without permission (Samuelson, 2023). This concern has been highlighted by the US Authors Guild, which has strongly opposed the unauthorized use of copyrighted materials by AI companies (USAuthorsGuild, 2023). To resolve such issues, the model owner need to either obtain permission or revise the training process and data (Friedman, 2025), highlighting the importance of offline support in mitigating LLM copyright risks.

### 3.2. Limitations of Offline-Only Approaches

Offline approaches address the infringement risks by labeling copyrighted data and updating the model through retraining or fine-tuning. While these methods are essential for long-term optimization, relying solely on them has limitations as well.

**Resource requirements and latency.** Implementing offline approaches often demands significant time and resource expenditures (Naveed et al., 2024; Zhang et al., 2023), such as retraining the model or conducting large-scale data cleansing. However, copyright concerns often require immediate handling. For instance, when a model generates sensitive or copyrighted content, online approaches can instantly detect and filter such output to prevent violations. In contrast, offline approaches are unable to provide such real-time intervention. If only offline approaches are applied to LLMs, any copyright violations can only be addressed after the offline approach is modified, implemented, and the model redeployed. This process can be time-consuming and resource-intensive. Such delays may allow frequent copyright infringements to go unchecked, thereby increasing the risk of legal disputes.

**Lack of dynamic adaptability to user behavior.** Unlike online approaches, offline methods do not directly influence real-time interactions between users and the model. This inherent limitation means that offline approaches lack the ability to dynamically adapt protective mechanisms based on the evolving context of user interactions. For example, consider scenarios where users input prompts specifically designed to exploit copyrighted material—such as those outlined by a recent study (Xu et al., 2024)—and request tasks like extraction, repetition, paraphrasing, or translation. In such cases, offline approaches will struggle to detect and prevent potential copyright violations in real-time. A primary reason for this is that offline approaches mainly intervene during model development and training, addressing patterns that are already identified. This restricts their ability to respond to new or unforeseen issues that arise during live usage. Therefore, combining them with dynamic, real-time online protection approaches is essential to effectively tackle these adaptability challenges.

**Difficulties in copyrighted dataset construction.** Many offline strategies, such as data cleaning and machine unlearning, often rely on removing or forgetting a specific copyrighted dataset. However, the difficulties of constructing such a comprehensive and accurate copyrighted dataset are currently overlooked. Take recent machine unlearning studies (Eldan & Russinovich, 2023; Yao et al., 2024) as an example; they often assume the presence of a predefined target dataset to be unlearned, such as books, blogs, or wiki-like entries related to *Harry Potter*. These studies only focus on the method itself, so it is reasonable to do so. However, in real-world scenarios, constructing a comprehensive unlearning dataset is far more complex than simply collecting data related to *Harry Potter*. Determining the appropriate contents to include in the copyrighted dataset and figuring out how to obtain them are both significant challenges (Cohen et al., 2024; Liu et al., 2025). An improperly constructed dataset can either compromise the utility of LLMs or result in inadequate copyright mitigation. We argue that achieving accurate and comprehensive dataset construction requires additional support to ensure both precision and completeness.

### 3.3. Necessity of Joint Optimization

Online approaches are typically proactive, focusing on real-time detection and prevention, whereas offline methods emphasize retrospective analysis and long-term optimization. However, as discussed in Section 3.1 and Section 3.2, independent optimizations of these anti-infringement approaches are insufficient to address the complexities of detecting and preventing LLM infringement. We observe that these ap-

proaches are complementary in nature; for example, online methods lack the ability to trace back to specific training phrases or datasets, an area where offline mitigation can provide critical support. Conversely, offline approaches cannot dynamically adapt to user behavior or mitigate the risks in real time, but online approaches can monitor and respond to user interactions as they occur, providing immediate mitigation. Therefore, to manage the copyright risks effectively and holistically, we argue that it is necessary to integrate online and offline approaches, optimizing the entire system by leveraging the strengths of both methods while compensating for their respective limitations. Such joint optimization can accommodate the complexity of LLM copyright issues and address the copyright challenges that emerge throughout the LLM lifecycle.

## 4. Alternative Views

We believe that copyright infringement issues, regardless of which stage they arise in the LLM lifecycle, should be properly addressed. However, some researchers argue that using copyrighted data for model training is reasonable, and that the primary focus of copyright risk should be on the outputs of LLMs rather than the training process. According to a recent study on copyright risks associated with LLMs (Rahman & Santacana, 2023), the authors contend that the definition of the "fair use" doctrine should primarily target the risks of LLMs producing regurgitated copyrighted material, rather than using the copyrighted content during the training process, as training may not inherently constitute a copyright violation. Furthermore, they suggest that tools to prevent copyright violations should prioritize developing mechanisms to detect instances where an LLM reproduces copyrighted content from its training data. Despite this viewpoint, the use of copyrighted data for training remains a highly contentious issue, with ongoing debates about its legality and ethical implications. Therefore, our proposal underscores the importance of addressing these concerns across the entire LLM lifecycle, especially in the absence of a clear consensus on whether copyrighted data can be ethically and legally used for training purposes.

Another perspective is that effectively addressing the copyright risks of LLMs primarily relies on the evolution and updating of legal frameworks. For instance, recent studies (Lucchi, 2024) have suggested strategies such as establishing clear data-sharing agreements, implementing compensation models like revenue sharing or royalties, and setting up data repositories or clearinghouses. Actually, recent legislation has already begun clarifying regulations for AI. In June 2024, the EU adopted the world's first set of AI rules, which include provisions on AI-related infringements (European Parliament, 2023), with some taking effect in February 2025. Additionally, extensive discussions on LLM copy-

right infringement legislation are ongoing (Cyphert, 2023; Ørstavik, 2025; Baack et al., 2025). While legal approaches are valuable for addressing the ethical and legal implications of LLMs, they often struggle to keep pace with rapid technological advancements, as a mature legal framework often takes decades to develop. The dynamic nature of AI development means that laws and regulations can quickly become outdated or insufficient. Therefore, we argue that a technology-based solution is needed to effectively manage copyright risks in the absence of a fully established legal framework.

## 5. Online-Offline Unified Framework to Manage Complex LLM Copyright Risks

By focusing on the inter-connections between the various copyright risks through LLM lifecycle and adopting a joint optimization approach, it is possible to leverage the strengths of both online and offline approaches while compensating for their respective limitations.

Based on this insight, we propose a generic **iterative online-offline joint optimization framework** to systematically mitigate LLM infringement risks. This framework ensures seamless online-offline coordination and iterative refinement for continuous infringement detection and prevention. This section starts with a comprehensive overview of the framework, followed by a detailed analysis of each component along with proposed new research directions. Finally, we will discuss how the framework can be integrated together to address LLM copyright risks more effectively.

### 5.1. Framework Overview

Figure 3 illustrates our proposed framework, which comprises three major parts: online mitigation, copyright examination, and offline calibration. The online mitigation module monitors and manages the model's real-time behavior to address immediate risks. The examination module bridges the mitigation and calibration stages, conducting comprehensive analyses and validation of potential infringement risks. Finally, the offline calibration module leverages the results of these analyses to improve various system components, including the training pipeline, mitigation strategies, and the model itself.

It is important to note that these modules are not working independently but integrated tightly, with continuous interaction and mutual reinforcement. This seamless integration is a key characteristic of our framework. The data collected during the model's online serving phase aids the examination process in identifying the root causes of infringement behaviors. These insights, in turn, inform the offline calibration process, which enhances the model's sensitivity to infringement issues throughout both training and serving

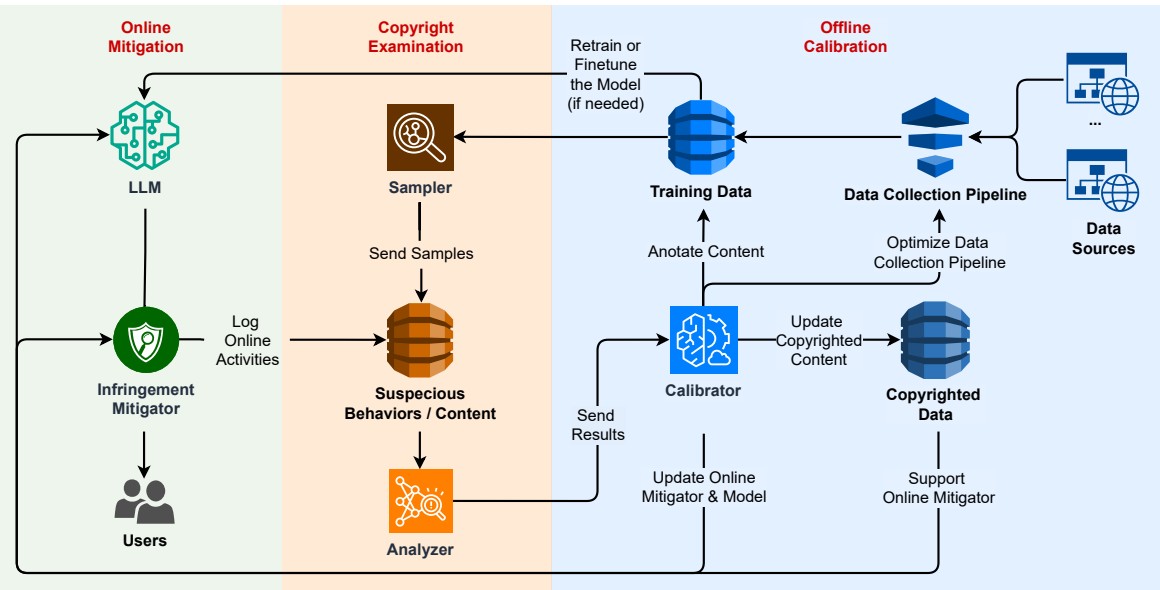

Figure 3. Online-offline unified framework to manage complex LLM copyright risks. This framework consists of three modules: online mitigation, copyright examination, and offline calibration. These modules work together through joint optimization to enhance the system's overall infringement mitigation capabilities.

stages. As a result, the model continually improves its performance in mitigating such issues over time. We provide an extra table to better illustrate the key components of the framework in Appendix B.

**Design principles.** (i) *Complementary:* Leverage the strengths of both online and offline methods to effectively address LLM copyright issues. (ii) *Comprehensive:* Address the root causes of copyright infringement across all stages of the LLM lifecycle—from data collection and model training to deployment and model updates, ensuring end-to-end protection. (iii) *Continuous Optimization:* Iteratively refine and dynamically update framework modules to adapt to evolving copyright risks.

### 5.2. Real-Time Online Infringement Mitigation

The online infringement mitigator is usually implemented in the intermediate layer between users and LLMs. This component is designed to monitor and manage the runtime behavior of the model in real time, enabling the timely detection and handling of potential infringement risks. Common approaches include blocking sensitive outputs or triggering alerts to inform users of potential issues. While significant research has already been conducted on the online mitigation phase, we believe that there remain numerous unexplored research opportunities in this area that are worth investigating further.

**Understand LLM's internal reasoning process.** Existing online infringement mitigation approaches rely primarily

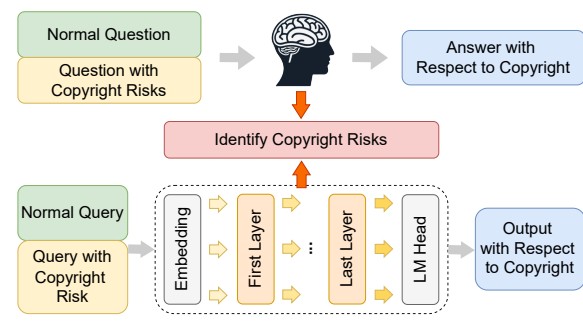

Figure 4. Identify copyright risks during internal reasoning process and prevent the risks before actual responding.

on system prompts, output filtering, or decoding-time interventions. However, none of these methods attempt to understand the underlying reasoning process that leads to potential copyright violations. We argue that understanding the model's internal reasoning process can enable more effective online mitigation of LLM generating copyrighted data. Recent research has already demonstrated the utility of analyzing the internal states of the model to detect certain behaviors, such as deception (Azaria & Mitchell, 2023) and hallucination (Ji et al., 2024). Building upon this line of work, we propose a new research direction: leveraging the internal states of LLMs as predictive signals for infringement behaviors.

Figure 4 illustrates this process. Just as humans can un-

derstand potential copyright risks through reasoning and ensure that their creations respect copyright, we believe it is also possible for LLMs to understand copyright risks in their internal states and identify potential issues even before decoding starts. Such copyright risk prediction mechanism based on the LLM internal state can offer deeper insights into the reasoning process behind harmful outputs and potentially achieve higher accuracy in detecting non-literal harmful content. Furthermore, compared to output filtering or decoding-time interventions, the approach based on LLM internal states can preemptively terminate the generation process even before decoding begins, resulting in improved computational efficiency.

**Contextual understanding.** Context-aware LLM behavior adjustment is another novel approach to explore under our framework. It can mitigates copyright risks by dynamically tailoring the model's responses based on the specific context of the input. This strategy involves assessing contextual factors such as the content type, user intent, and potential copyright implications before generating an output. For instance, if the input query involves highly sensitive or identifiable copyrighted content, the model should prioritize safer responses, such as providing general information or redirecting the user to licensed sources. Some studies have explored context-aware LLM behavior adjustment (Luu et al., 2024; Kannadasan, 2024), but research on applying it to address LLM infringement issues remains limited.

**Log suspicious activities.** Current approaches usually trigger infringement prevention strategies when the detected infringement risk exceeds a predefined *risk threshold.* However, variations in models, regional copyright regulations, and user demographics can all lead to fluctuations in these *risk thresholds.* Taking methods like MemFree (Ippolito et al., 2023) and SHEILD (Liu et al., 2024a) as examples, they rely heavily on the selection of the n-gram length. If $n$ is set too low, the mitigator achieves high recall but suffers from very low precision, leading to over-filtering issues. Conversely, if $n$ is set too high, the mitigator's precision improves, but its recall drops significantly, resulting in under-filtering problems. Given the complexity of real-world environments, scenarios where the risk exceeds the threshold may not always indicate actual infringement, and cases below the threshold may still involve potential violations. We argue that the context of such potential risks and the decision-making process of the mitigator should be accurately logged under appropriate circumstances. In our proposed framework, these logs can then serve as input for subsequent copyright examination processes, enabling comprehensive analysis of violations and root cause tracing.

**Awareness of localized copyright status.** In response to Section 3.1, we also want to emphasize the importance of the online mitigator's awareness of localized copyright regulations. This can be achieved by detecting the user's IP location and cross-referencing it with local copyright databases. It is important to note that these localized copyright datasets are dynamic and constantly evolving. To ensure the online mitigator performs effectively, an offline calibration process is essential to maintain and update a localized copyright dataset. This need for continuous adaptation is a key motivation behind our proposed online-offline joint optimization framework. More details will be discussed in Section 5.4.

### 5.3. Copyright Examination

The copyright examination module serves as a bridge between online mitigation and offline calibration. On one hand, it processes the data logged during the online mitigation process; on the other hand, it extracts potentially copyright-sensitive content from the offline database for in-depth analysis. As a centralized data processing module within our framework, it efficiently integrates online monitoring with offline calibration, enhancing the framework's capability and effectiveness in addressing copyright infringement risks.

**Analyze Online activities.** The online activities include user prompts, model outputs, the triggered online mitigator actions, and other real-time metrics. By analyzing user behavior, the system can identify prompts with high copyright risk. This is particularly valuable as a recent study revealed that most LLMs fail to adequately respect copyright-sensitive information in user prompts (Xu et al., 2024). Additionally, analyzing the model's actual outputs and the mitigator's actions enables the system to assess whether the mitigator suffers from over-filtering or under-filtering issues. By examining other real-time performance metrics, such as filtering latency and trigger frequency, we can gain deeper insights into the impact of online mitigation on the overall performance of the model. These findings can provide a more accurate and data-driven basis for optimizing the online mitigation process.

**Audit samples from training data.** As discussed in Section 3, a key challenge in managing LLM copyright risks stems from the dynamic nature of copyrighted content and the diverse legal regulations across different regions. In our framework, this issue is effectively addressed through training data sampling and analysis during the copyright examination stage. Copyright-related data will be sampled from the training data and validated using manual or algorithmic approaches (Pan et al., 2025), incorporating region-specific legal provisions to assess potential copyright infringement within particular legal requirements. This process enables the identification of additional infringing content and facilitates the proactive expansion of the copyright database. Appendix A contains some public copyright status databases that can serve as a starting point for this work.

## 5.4. Systematically Offline Calibration

The offline calibration stage leverages insights from the examination stage to enhance the system's overall ability to address copyright issues comprehensively. This is achieved by maintaining and updating the copyrighted dataset, refining online mitigation strategies, updating the model itself, and optimizing the LLM training pipeline. This process not only resolves identified issues but also establishes a more robust and efficient prevention mechanism to mitigate potential infringement risks in the future.

**Maintain localized copyrighted dataset.** First of all, the offline calibrator is responsible for maintaining a dynamic, localized copyrighted dataset that is continuously updated with the latest copyrighted data in accordance with specific regulatory requirements of different countries and regions. As mentioned in Section 3, many online mitigation strategies and offline optimization methods greatly benefit from this dataset. Online copyright infringement mitigation approaches, such as MemFree (Ippolito et al., 2023) and SHIELD (Liu et al., 2024a), heavily rely on accurate copyrighted databases as their ground truth. Similarly, offline methods like machine unlearning and data cleaning also depend on these copyrighted datasets to provide essential guidance.

**Optimize online mitigation strategy.** After understanding the copyright risks that the model faces in the actual serving stage, the calibrator will then be able to implement more sophisticated mitigation strategies to update the online mitigator. This includes specifying more precise output filtering parameters for online mitigation algorithms such as MemFree (Ippolito et al., 2023). The calibrator can also deploy region-specific mitigation strategies based on local legal requirements. Furthermore, by integrating real-time feedback from the model, the calibrator can dynamically adjust the level of governance. Through this approach, the framework can effectively reduce false positive rates and enhance the response time of the online mitigator, striking a balance between ensuring system compliance and maintaining a seamless user experience.

**Update the model.** The offline calibrator is also responsible for updating the model if needed. As analyzed in Section 3.2, one of the key challenges faced by existing methods in updating models is the reliance on a well-defined copyrighted dataset for guidance. Our framework addresses this issue by dynamically maintaining a comprehensive copyrighted dataset, thereby simplifying the model update process. Several existing techniques can be seamlessly integrated into our framework. For instance, machine unlearning techniques (Eldan & Russinovich, 2023; Yao et al., 2024) can be applied more effectively to make the model forget specific copyrighted data, and other fine-tuning methods (Singh et al., 2024) can be leveraged to enhance data confidentiality.

**Optimize the LLM training cycle.** Based on the analysis results from the copyright examination phase, the offline calibrator can further optimize the entire LLM training cycle, ensuring that future models exhibit better awareness of copyright issues. The calibrator can trace newly identified copyrighted data back to the corresponding data collection pipeline and take proactive measures. For instance, it can adjust the data collection pipeline to bypass certain copyright-protected sources or annotate copyright information during the data collection phase. In the training process, existing research has shown that embedding *copyright watermarks* into training data can assist in detecting copyrighted content (Wei et al., 2024b). These measures establish a solid foundation for developing more responsible and sustainable LLMs, enhancing their capacity to effectively address copyright-related concerns.

## 5.5. Iterative Online-Offline Joint Optimization

The proposed iterative online-offline joint optimization framework achieves dynamic management of copyright risks through the close collaboration of three core components. The online mitigation module monitors and handles potential infringement risks in real time; the copyright examination module systematically analyzes and understands the collected data; and the offline calibration module optimizes the system based on the analysis results.

**Feedback mechanism.** The feedback mechanism ensures a synergistic interplay between the online and offline modules. Specifically, the data collected by the online module helps to identify the specific manifestations and root causes of copyright risks, thereby providing precise directions for the offline optimization process. In turn, the offline module maintains a localized copyrighted dataset, refines mitigation strategies, updates the model, and enhances other system components. These refinements collectively reduce the copyright risks encountered by the LLM during the online serving stage.

**Iterative process.** It is worth noting that the proposed framework contains an iterative process that can achieve continuous improvement in copyright risk management capabilities. Each round of iteration not only improves the model's recognition accuracy for known infringement patterns, but also enhances its adaptability to new types of infringement. As the number of iterations increases, the system accumulates more empirical data, making the risk prevention mechanism more complete and the model's behavior more controllable. This gradual optimization process ensures that the system can continue to adapt to the evolving copyright environment and minimize potential risks.

**Balance between online and offline approaches.** This joint optimization framework allows online and offline anti-infringement approaches to complement each other, achiev-

ing the level of efficiency and effectiveness that neither approach could attain independently. The online module focuses on rapid response and timely intervention, identifying and addressing potential infringement risks within milliseconds. Meanwhile, the offline module leverages its computing power and access to historical data for in-depth pattern mining and long-term optimization. This design ensures real-time protection while also driving continuous performance improvement over time.

This iterative online-offline joint optimization can be flexibly implemented based on practical needs. Additional discussions can be found in Appendix C and Appendix D.

## 6. Conclusion

This position paper argues that effectively managing LLM copyright risks necessitates an iterative online-offline joint optimization strategy. We analyze the complexity of LLM copyright infringement challenges and highlight the limitations of current mitigation methods, emphasizing that standalone measures are insufficient to address the evolving risks comprehensively. To bridge this gap, we propose a novel, dynamic and systematic framework that seamlessly integrates online and offline approaches, tackling copyright risks across all stages of the LLM lifecycle while iteratively adapting to emerging challenges. Additionally, our analysis and framework offer valuable insights for potential research directions and shaping legal regulations. Future research should explore the new directions proposed in this paper while considering the interconnected nature of LLM copyright issues. This will enable the development of comprehensive strategies to address potential copyright concerns that may arise at various stages of the LLM lifecycle.

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

## A. Existing Copyright Status Database

There are some existing official copyright status databases operated or supported by national authorities or public institutions, as shown in in Table 2.

*Table 2.* Selected existing official copyright status databases

| Country | Database | Reference |
|---------|----------|-----------|
| USA | U.S. Copyright Office Public Records | (U.S. Copyright Office, 2025) |
| Canada | Canadian Copyright Database | (Canadian Intellectual Property Office, 2025) |
| Australia | National Library of Australia | (National Library of Australia, 2025) |
| China | National Works Registration Database | (China Copyright Protection Center, 2025) |

## B. Key Components for the Online-Offline Unified Framework

The following Table 3 provides an overview of the key components involved in the proposed framework. Each component's role, access requirements, associated costs, and workflow are outlined to provide a clear understanding of how they interact in the overall process.

*Table 3.* Overview of Key Components for the Online-Offline Unified Framework

| Component | Role | Access | Cost | Workflow |
|-----------|------|--------|------|----------|
| Infringement Mitigator | Monitors and manages the model's real-time behavior, logs potential risks | Access to real-time LLM interactions, flagged cases, and copyright status. | Low – real-time and automated. | Sends detected cases to the Analyzer for further inspection. |
| Analyzer | Examines flagged content for infringement risks and assesses its validity. | Access to model output logs and training databases. | Moderate – automated with potential human reviews. | Bridges Mitigator and Calibrator, analyzing issues from both the online and training phases. |
| Calibrator | Leverages insights from the examination stage to enhance the system's overall copyright awareness. | Access to training data and assessment results from the Analyzer. | It depends – minor updates are low-cost, retraining/pipeline modification could be costly. | Updates other components in the system, including infringement strategies, data processing pipelines, and the model itself. |

## C. Implementation Complexity

The proposed framework offers a flexible approach to managing copyright risks associated with LLMs, with the complexity of its implementation varying depending on the specific context. Below, we outline the considerations for different types of entities:

- **For Institutions Providing Large Foundation Models:** These entities typically already have established mechanisms in place to address copyright infringement. In this context, our framework serves as an enhancement, offering a structured approach to further strengthen existing copyright compliance practices and mitigate risks more effectively.

- **For Small Teams with Limited Resources:** For smaller teams, fully implementing the framework may pose challenges due to resource constraints. However, these teams can still effectively leverage the framework. (1) Small teams often work with narrow datasets and specific tasks, which simplifies the copyright compliance process. For example, a team developing a virtual LLM tutor may concentrate on a limited set of copyrighted textbooks, thereby reducing the scope of copyright concerns. (2 )The use of low-cost training and fine-tuning techniques [1-3] can further enhance the feasibility of implementing the framework in resource-limited settings.

- **For the LLM Community:** Community members can contribute to various components of the framework. Contributions may include the creation and maintenance of shared copyright databases, the development of open-source algorithms for copyright infringement mitigation.

## D. Mapping Key Framework Components to the LLM Lifecycle

Figure 5 maps the key components of our framework to the LLM lifecycle. It shows the logical connection between Figure 1 (LLM lifecycle and potential copyright risks) and Figure 3 (detailed workflow of the proposed framework).

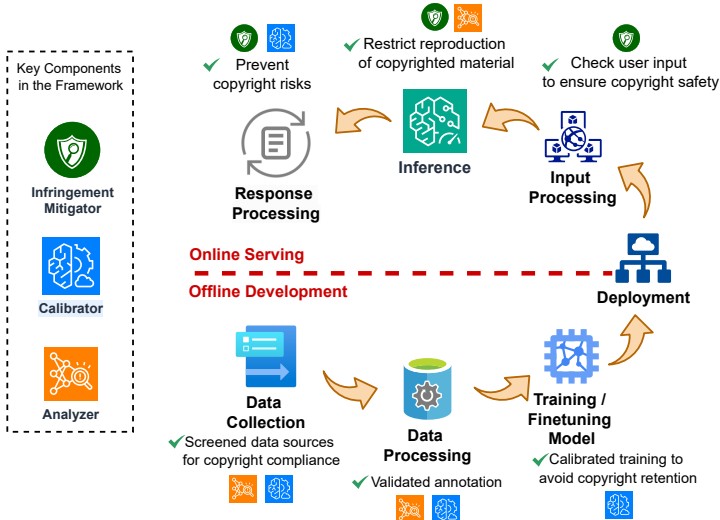

*Figure 5.* Component-wise mapping of the proposed framework to each stage of the LLM lifecycle.

