# OpenReview forum: "Position: Iterative Online-Offline Joint Optimization is Needed to Manage Complex LLM Copyright Risks"
_ICML.cc/2025/Position_Paper_Track — ICML 2025 Position Paper Track poster_

### Official Review · Reviewer_xwHU · 2025-03-06

**Significance:** 3
**Argument Clarity:** 3
**Rating:** 3
**Confidence:** 4

**Questions:**

NA

**Discussion Potential:**

3

**Paper Summary:**

The paper deals with an important and timely problem, the copyright issue in the LLM. It shows the limitation of current existing methids and introduce a new framework. The paper is more likely to a survey paper, which concludes the known methods and proposes new insights.

**Position:**

Yes

**Position In Title:**

Yes

**Related Work:**

3

**Strengths And Weaknesses:**

Pros:
1. A timely and important problem is discussed.
2. The paper is good written.


Cons:
From my point of view, as a paper discussing the copyright issue in LLM, it should first give a formal definition or scope for what to discuss, for example, currently, we are still unknown about whether and how LLM constitutes copyright infringement. As you discuss, some researchers believe some use of copyrighted material belongs to fair use and therefore not considered as copyright infringement. Also, currently, there is no legal file that focuses on the copyright issue in LLM, can be refered. As a result, what I think to be more reasonable is focusing on the “copy” issues instead of the copyright issue. That means, what we want to check is whether the generated contents of LLM replicates the known texts. There are many interesting research points, here. For example, (1) how to define “copy”? (2) how much “copy” is safe? (3) whether LLMs replicate contents of known texts rather than generating new ones?

**Support:**

3

---

> ### Author Rebuttal · Authors · 2025-04-01
>
> ### [W1: Add a formal scope definition for what to discuss - R1: Sure!]
> > As a paper discussing the copyright issue in LLM, it should first give a formal definition or scope for what to discuss.
>
> Thanks for pointing this out, based on the guidance from U.S. Government Copyright Office [1,2], we provide the **Scope Definition** of our work: This paper focuses on the copyright law and policy issues raised by LLM, including the scope of copyright in LLM-generated works and the use of copyrighted materials in LLM training. We comprehensively analyze the complexity of copyright issues encountered by LLM, propose a new position and introduce a unified joint-optimization framework to manage LLM copyright risks.
>
> We have included the above definition into Section 1 and cited the official documents from U.S. Copyright Office Website to provide a clear scope of our discussion.
>
> ### [W2: It's still unknown about whether and how LLM constitutes copyright infringement - R2: Some law provisions already taken effect in Feb. 2025.]
> > Currently, we are still unknown about whether and how LLM constitutes copyright infringement. As you discuss, some researchers believe some use of copyrighted material belongs to fair use and therefore not considered as copyright infringement.
>
> We appreciate the reviewer for raising this important point. To clarify:
> 1. **Existing Law Framework:** In fact, latest laws have already established clearer regulations for AI. In June 2024, the EU adopted the world’s first set of AI rules, which include provisions on AI-related infringements [3]. Some of these provisions have already taken effect starting in Feb. 2025. Additionally, extensive discussions on LLM copyright infringement legislation are ongoing, as seen in [4-6]. We have included them as related work in Section 4.
> 2. **Ongoing lawsuits:** There are numerous lawsuits regarding LLM copyright infringement that have already consumed significant money and resources [7,8], making this a critical issue worthy of attention.
> 3. **Updates in the paper:** We revised multiple expressions in our paper to avoid potential misunderstandings. For instance, we have replaced "LLM copyright infringement" in Section 5 with "LLM generating copyrighted data".
>
> ### [W3: Focus on the “copy” issues instead of the "copyright" issue - R3: Please see our clarification]
> > What I think to be more reasonable is focusing on the “copy” issues instead of the copyright issue. That means, what we want to check is whether the generated contents of LLM replicates the known texts. There are many interesting research points, here. For example, (1) how to define “copy”? (2) how much “copy” is safe? (3) whether LLMs replicate contents of known texts rather than generating new ones?
>
> Thanks for the thoughtful insights, to clarify:
> - **Key topic of ICML2025:** The "**concerns about data legality and copyright**" is one of the key topics listed in the [ICML 2025 call](https://icml.cc/Conferences/2025/CallForPositionPapers), we greatly value this opportunity to spark more discussion within the LLM community regarding this topic.
> - **Emerging field with insufficient discussion:** As we noted in our response to W2, legal constraints on LLM copyright risks are becoming increasingly clear. The LLM community should have more comprehensive discussions on this issue, which is one of the motivations of our work.
> - **Distinct challenges**: the LLM copyright risks present many distinct **legal concerns** and **technical challenges** that are not present in the general "copy issues", such as detecting copyrighted content in model outputs and tracking the evolving copyright status over time and across jurisdictions.
>
> We kindly request a further clarification from the reviewer if the reviewer still suggest to **NOT** focus on "copyright" concerns?
>
>
> **References (simplified due to word limitation)**
>
> [1] U.S. Copyright Office, "Copyright Registration Guidance for Works Containing AI-Generated Materials"
>
> [2] U.S. Copyright Office, "A REPORT of the register of copyrights"
>
> [3] European Parliament Official Website, "EU AI Act"
>
> [4] Inger Berg Ørstavik, "Development of Large Language Models: Copyright Law Perspectives for Research Institutions and Research Libraries"
>
> [5] Amy B. Cyphert, "Generative AI, Plagiarism, and Copyright Infringement in Legal Documents"
>
> [6] Stefan Baack, et al. "Towards Best Practices for Open Datasets for LLM Training"
>
> [7] Joshua Freeman, et al. "Exploring Memorization and Copyright Violation in Frontier LLMs: A Study of the New York Times v. OpenAI 2023 Lawsuit"
>
> [8] The Authors Guild, "More than 15,000 Authors Sign Authors Guild Letter Calling on AI Industry Leaders to Protect Writers"

---

> > ### Comment · Reviewer_xwHU · 2025-04-01
> >
> > Thanks author's rebuttal. Could I ask whether for the position track, there is no need for experiments?
> >
> > I am sorry for not familiar with this track. I have raised the score to 3.

---

> > > ### Author Response · Authors · 2025-04-02
> > >
> > > Thanks to the reviewer for the timely reply!
> > >
> > > According to the official website of ICML 2025, position papers present an argument for a viewpoint or perspective on **what should be done**, in contrast to research track papers, which report on completed advancements. Therefore, experiments are not required. More details can be found at [ICML 2025 Position Paper Offical Website](https://icml.cc/Conferences/2025/CallForPositionPapers).

---

### Official Review · Reviewer_nmAn · 2025-03-13

**Significance:** 4
**Argument Clarity:** 3
**Rating:** 5
**Confidence:** 4

**Questions:**

What are the examples that best illustrate the various approaches that you describe, and that would support your statement regarding the need for a joint optimized framework?

**Discussion Potential:**

4

**Paper Summary:**

The paper concerns copyright challenges related to LLM development and argues for a systemic approach that combines several existing approaches to risk management. The paper distinguishes three types of approaches: based on mitigation, examination and calibration, and begins with an overview of approaches classified as either online or offline. The authors then argue for join optimization of approaches which they consider complementary. The second part of the paper proposes such a framework, which allows systematic mitigation of LLM infringement risks and includes mitigation, examination and callibration approaches. The framework is iterative, with three main components collaborating together.

**Position:**

Yes

**Position In Title:**

Yes

**Related Work:**

4

**Strengths And Weaknesses:**

The paper offers an interesting approach that takes into account several different methods of mitigating copyright challenges in LLM development. Based on this, it offers a systemic framework, arguing that such an approach avoids the risk inherent to more specific approaches. This framework, including the online / offline distinction and classification of various approaches offers not only a proposal for a unified approach, but an interesting classification of approaches to copyright-related risk in LLm development. As such, it offers an opportunity for interesting and important conversations related to datasets and copyrights in them.
The paper is relatively clearly argued, but the argument is very dense and sometimes difficult to parse, as it describes a quite complex framework of various approaches that are described in an abstract way.

**Support:**

4

---

> ### Author Rebuttal · Authors · 2025-04-01
>
> ### [W1: The argument is dense and sometimes difficult to parse - R1: Please see our clarifications]
> >The paper is relatively clearly argued, but the argument is very dense and sometimes difficult to parse, as it describes a quite complex framework of various approaches that are described in an abstract way.
>
> Thank you for highlighting this concern! The complexity of our framework is inherent to the complexity of copyright challenges throughout LLM's lifecycle. To better illustrate the components of our framework and improve clarity, we have added a detailed table with examples in the Appendix. Specifically:
>
> |Key Component|Role          |Workflow    |
> |-------------|--------------|------------|
> | **Infringement Mitigator** | Monitors and manages the model’s real-time behavior, logs potential risk|Sends detected cases to the **Analyzer** for further inspection.|
> |**Analyzer**|Examines flagged content for infringement risks and assesses its validity.|Bridges **Mitigator** and **Calibrator**, analyzing issues from both the online and training phases |
> | **Calibrator** | Leverages insights from the examination stage to enhance the system’s overall copyright awareness |  Updates other components in the system, including the infringement strategies, the data processing pipelines and the model itself|
>
>
>
> ### [Q1: Key examples to illustrate the approaches? - A1: Sure!]
> > What are the examples that best illustrate the various approaches that you describe, and that would support your statement regarding the need for a joint optimized framework?
>
> Thank you for the question. Here are some key examples to illustrate the need for the joint optimized framework:
> - **Localized copyright status.** Some works may be public in certain countries but copyrighted in others. Recent research [1] shows that current LLMs and online mitigations fail to handle this mixed copyright status issue. To address this, our proposed framework introduces a dynamic, region-specific copyright database. Such database facilitates an online mitigator that monitors and manages LLM's interactions, ensuring compliance with localized copyright restrictions.
> - **Non-literal copying** The rephrases or paraphrases content, altering its wording or structure while preserving the original meaning, which still raises copyright concerns. Such conceptual similarities cannot be fully detected by online methods, which emphasizes the need for offline approaches such as model fine-tuning or adjustments to training data to address such issues.
> - **User Input Variability**: User input is highly dynamic and unpredictable. Some users may deliberately craft prompts to bypass copyright restrictions, such as requesting copyrighted content in multiple languages[2]. Relying solely on offline methods like fine-tuning or retraining is impractical in such cases, as user inputs frequently change. Offline approaches also come with high costs and administrative limitations. Therefore, involving flexible online detection and intervention strategies is essential to address these challenges effectively.
>
> We emphasized the above examples in our paper to improve our story telling and better illustrate the need for the proposed framework.
>
> **References**
>
> [1] Liu, Xiaoze, et al. "Shield: Evaluation and defense strategies for copyright compliance in llm text generation." arXiv preprint arXiv:2406.12975 (2024).
>
> [2] Chen, Yupeng, et al. "Beyond English: Unveiling Multilingual Bias in LLM Copyright Compliance." arXiv preprint arXiv:2503.05713 (2025).

---

### Official Review · Reviewer_1ebW · 2025-03-15

**Significance:** 3
**Argument Clarity:** 4
**Rating:** 4
**Confidence:** 3

**Questions:**

(1) Could non-literal copying be mitigated by performing a neighborhood search in terms of the prompt's semantics?

(2) Figure 3 introduces several new parties, including the calibrator, analyzer, and infringement mitigator. Could the authors briefly discuss how these parties might operate in practice? Specifically, it would be valuable to understand their intended roles, what access or knowledge each would require, the potential costs for their operation, and how they would collaborate within the proposed framework.

**Discussion Potential:**

3

**Paper Summary:**

This paper examines the critical challenge of addressing copyright issues in large language models (LLMs). The authors offer a valuable taxonomy that classifies existing mitigation methods into two categories: online approaches (implemented after deployment) and offline approaches (implemented before deployment). Through careful analysis, they convincingly demonstrate that neither approach alone provides a comprehensive solution, as each has inherent limitations that may lead to failure under specific circumstances. To overcome these shortcomings, the paper proposes an innovative online-offline unified framework and introduces novel mitigation strategies that show promise for more effectively addressing LLM copyright concerns.

**Position:**

Yes

**Position In Title:**

Yes

**Related Work:**

3

**Strengths And Weaknesses:**

### Strengths
Overall, I thoroughly enjoyed reading this paper, which addresses an important and complex problem in a timely manner. I am particularly impressed by:
- The clear classification of existing mitigation methods
- The identification of pitfalls in existing approaches, revealing critical gaps in current solutions
- The legal and policy perspectives on LLM copyright issues
- The convincing unified framework that effectively leverages existing methods while proposing new solutions across the complete lifecycle of LLMs

### Weaknesses
- **Section 3, localized copyright status**: This problem appears to be primarily a factual issue caused by LLMs (and likely represents an unsuccessful online approach) that could potentially be addressed by an improved online approach alone. While the authors' proposed solution based on "awareness of localized copyright status" is conceptually sound, they could provide stronger justification for why this necessitates their unified framework rather than an enhanced online-only solution. Furthermore, it remains unclear whether this is an isolated example or a frequently occurring issue. The authors would strengthen their case by providing additional real-world examples of such localized copyright conflicts to demonstrate the scope and significance of this particular challenge.
- **Figure 3**: I believe Figures 1 and 3 lack visual cohesion in terms of style and arrangement. The reader's comprehension would be significantly enhanced if Figure 3 were redesigned to visually extend from the classification framework established in Figure 1.

minor: the heading of each page should be changed to the title.

**Support:**

3

---

> ### Author Rebuttal · Authors · 2025-04-01
>
> ### [W1: More examples to demonstrate the significance of the localized copyright issues - R1: Sure!]
> >Section 3, localized copyright status: ... the author could provide stronger justification for why this necessitates their unified framework ... it remains unclear whether this is an isolated example or a frequently occurring issue...
>
> Thanks for the thoughtful comments, to clarify on this:
>
> The localized copyright issue mentioned in our paper is not an isolated example but aligns with a lot of existing research. For instance, the recent study[1] from Purdue University that we cited in Section 2 specifically constructed a "Partially Copyrighted" dataset called "BS-PC," where the content is copyrighted in some countries but not in others. They used this dataset to evaluate various online mitigation methods and multiple mainstream LLMs. Their results clearly indicate that existing online-only methods are insufficient to address the complexities of this issue, whereas our proposed unified framework provides a structured approach to handling it.
>
> To emphasize the prevalence of localized copyright issues, we have incorporated the above discussion into Section 3.
>
> ### [W2: Improve the visual design of the Figures - R2: Sure!]
> > Figures 1 and 3 lack visual cohesion in terms of style and arrangement...
>
> Thank you for your valuable suggestion! To clarify: figure 1 provides a high-level overview of the LLM lifecycle, highlighting various potential copyright risks. In contrast, Figure 3 presents very detailed workflow of the proposed framework. While these two figures are closely related, their visual differences stem from their distinct purposes and perspectives.
>
> To address this issue, we have added a new figure in the appendix, mapping the key components of our framework onto the cycle in Figure 1. Please check the new figure through this [anonymous figure](https://anonymous.4open.science/r/ICML2025Rebuttal-3FE3/README.md).
>
>
> ### [Q1: Semantical neighborhood search to mitigate non-literal copyright risks? - A1: Please see our clarification]
> >Could non-literal copying be mitigated by performing a neighborhood search in terms of the prompt's semantics?
>
> Thanks to the reviewer for sharing the ideas! From the technical perspective, the key challenge in detecting non-literal infringement lies in the fact that infringers often avoid direct duplication, instead employing different vocabulary or structural modifications to convey the same ideas. We agree that semantic search can be useful in identifying plagiarism through paraphrasing, synonym substitution, reordering, and other obfuscation techniques, making it a potentially valuable tool for detecting non-literal infringement.
>
> To make our paper more comprehensive, we have incorporated the above discussion into Section 3.
>
> ### [Q2: Add detailed discussions on the system components? - A2: Sure!]
> >Figure 3 introduces several new parties... it would be valuable to understand their intended roles, what access or knowledge each would require, the potential costs for their operation, and how they would collaborate within the proposed framework.
>
> Thank you for your insightful comments. Below is a breakdown of the key components in our framework, we've added it into the appendix of our paper:
>
> | Component| Role | Access | Cost | Workflow |
> |-|-|-|-|-|
> | **Infringement Mitigator** | Monitors and manages the model’s real-time behavior, log potential risks | Access to real-time LLM interactions, flagged cases and copyright status. | Low – realtime and automated. | Sends detected cases to the **Analyzer** for further inspection. |
> | **Analyzer** | Examines flagged content for infringement risks and assesses its validity. | Access to model output logs and training databases. | Moderate – automated with potential human reviews. | Bridges **Mitigator** and **Calibrator**, analyzing issues from both the online and training phases |
> | **Calibrator** | Leverages insights from the examination stage to enhance the system’s overall copyright awareness | Access to training data and assessment results from **Analyzer**. | It depends – minor updates are low-cost, retraining/pineline modification could be costly. | Updates other components in the system, including the infringement strategies, the data processing pipelines and the model itself.|
>
> **References**
>
> [1] Liu, Xiaoze, et al. "Shield: Evaluation and defense strategies for copyright compliance in llm text generation." arXiv preprint arXiv:2406.12975 (2024).

---

> > ### Comment · Reviewer_1ebW · 2025-04-07
> >
> > Thank you for the rebuttal. The new figure looks great and I appreciate the further discussions.

---

### Official Review · Reviewer_tUt3 · 2025-03-26

**Significance:** 3
**Argument Clarity:** 3
**Rating:** 3
**Confidence:** 4

**Questions:**

1. Do subproblems like unlearning pose a significant challenge because they are an active area of research but also hard to quantify and implement?
2. How will this research be doable by research teams that have limited resources or that do not train their own base models? Does the requirement to have an online mitigation layer make this research hard to do for researchers that are not building their own LLMs?
3. Are there any well known copyright datasets (with information like location specific rules) that are publicly available? Are these datasets often not available because of risk of copyright infringement? Does that mean this research can only be done by larger organizations that acquire such information from other organizations in bulk (e.g., NYTimes articles, book texts, etc.)
4. How much does the proposed framework rely on having an understanding of Internal reasoning of an LLM? Given that it is a hard problem to solve, what are the alternatives?

**Discussion Potential:**

4

**Paper Summary:**

The paper advocates for a joint optimization framework to handle copyright risks in LLMs, and strongly argues that individually enforcing online or offline mitigation strategies to handle copyright risks is inadequate because the risks arise at various lifestages of an LLM including data collection/preprocessing, training, finetuning, deployment, inference etc. While offline calibration/methods try to enforce it at the data collection, preprocessing and training level, the online methods need to be real time or quick so they act at the inference stage (either input/prompt processing or output processing levels). Authors provide examples at both these stages that demonstrate inadequacy of handling the problem at either level.
The proposal essentially is to introduce a "copyright examination" layer as a bridge between online mitigation and offline calibration to harness the data collected by the online mitigation layer that already identifies high-risk prompts and logs the actions of the online mitigator. The authors argue that a continuous iterative processing (like a feedback loop) that uses the data from online mitigation to inform offline calibration to improve a model's ability to mitigate risks online, hopefully improving them with each iteration.

**Position:**

Yes

**Position In Title:**

Yes

**Related Work:**

3

**Strengths And Weaknesses:**

Strengths:
1. The paper very clearly articulates the challenges arising due to solving copyright risks separately through the online and offline formats. This is a significant challenge, not only for the copyright risks (i.e., risk that the model will output verbatim content from copyrighted material), but also for safety risks attached with the model. The isolation of offline and online mitigation methods is also how many large organizations/companies structure their teams to try to solve the problem. This paper essentially points out the weaknesses of such approaches even at the organization level.
2. The proposal to continuously adapt the model to risks that arise during online mitigation and enforcement is a significant contribution of the paper and authors give a comprehensive framework to solving this problem by defining a feedback loop and identifying tasks like unlearning, data cleanup, finetuning, etc.

Weaknesses:
1. Implementation Complexity and requirements: While comprehensive, the framework requires access to the entire system of pre-training, post-training and inference/serving of an LLM. This is quite hard to have unless the LLM and the system are severely scaled down from the scale of the state of the art systems, or unless you are one of the few places with enough critical number of users and access to your own model training stacks. In other words, any meaningful instantiation of this framework would require significant resources and coordination which is unlikely to happen in most research teams.
2. "Internal Reasoning" of LLMs: the authors claim that understanding an LLM's internal reasoning for copyright infringement is a promising direction. However, this is a complex problem with very little success.
3. Copyright databases: Creating and continuously updating copyright datasets (as proposed to be used by examination and calibration modules) is an enormous task. It would be interesting to see if there are any attempts to maintain such datasets in the open source community or evidence of such efforts by teams building proprietary models. Perhaps, this is done more carefully in LLMs that generate code.

**Support:**

4

---

> ### Author Rebuttal · Authors · 2025-04-01
>
> ### [W1&Q2: Implementation Complexity?]
> > W1: While comprehensive, the framework requires access to the entire system of pre-training...
> > Q2: How will this research be doable by research teams that have limited resources ...
>
> Thanks for the valuable insights! Our proposed framework offers a flexible approach to managing LLM copyright risks, with implementation complexity varying by context. Specifically:
> - **For institutions that provide large foundation models:** These entities typically already have copyright infringement mechanisms in place, and our framework can help them improve their efforts.
> - **For small teams with limited resources:** Fully implementing our framework can be challenging, but small teams can still adapt it effectively. (1)Small teams usually focus on specific tasks and small datasets, which simplify copyright compliance and implementation. For instance, a virtual LLM tutor team might only focus on a small set of copyrighted textbooks. (2) Low-cost training and fine-tuning methods [1-3] also aid feasibility.
> - **For the LLM community:** Community members can contribute to various components of the framework, such as shared copyright databases or open-source infringement mitigation algorithms.
>
> We have included the above discussions in the appendix to clarify the paper's potential use cases in various groups.
>
> ### [W2&Q4: Complexity of the LLM's internal reasoning?]
> >W2: Understanding LLM's internal reasoning is a complex problem with very little success.
> >Q4: How much does the proposed framework rely on having an understanding LLM's internal reasoning? ... what are the alternatives?
>
> Thanks for raising this important concern. To clarify:
> - **Success cases:** As discussed in Section 5.2, many prior work has successfully used internal states to predict LLM behaviors (e.g., deception[4], hallucination[5,6], unseen query[5]), showing that meaningful signals can be extracted from the LLM internal reasoning stages.
> - **Flexibility of the framework:** Our framework is independent of the understanding of internal reasoning. Other online mitigation methods, such as prompting, filtering, and decoding-time techniques, serve as alternative solutions.
>
> ### [W3&Q3: existing copyright databases and challenges in maintaining copyright databases?]
> >W3: Creating and continuously updating copyright datasets ...  by teams building proprietary models.
> >Q3: Are there any well known copyright datasets? ...
>
> Creating and continuously updating comprehensive copyright datasets is indeed a significant challenge. To our knowledge, there isn't yet a well-established dynamic copyrighted database. Raising discussions about this gap is a key motivation of our paper.
>
> We did identified some existing copyright-related databases and have included them in our appendix:
> - **Copyright status database** maintained by national governments, for example:
>   - [U.S. Copyright Office Public Records](https://www.copyright.gov/public-records/)
>   - [Canadian Copyright Database](https://www.ic.gc.ca/app/opic-cipo/cpyrghts/)
>   - [National Library of Australia](https://catalogue.nla.gov.au/)
> - **Copyrighted content dataset for research**: Some benchmark research projects, like Copybench[7], have created and shared processed copyrighted content datasets for research.
>
> ### [Q1: Challenges of machine unlearning?]
> >Do subproblems like unlearning pose a significant challenge ... hard to quantify and implement?
>
> Machine unlearning is a promising approach to handle LLM copyright issues. To address your questions:
> - **Quantification**: Various unlearning evaluation metrics exist in prior research. [8] defines forget-quality metrics for bias removal, confusion resolution, and privacy, proving unlearning can be meaningfully quantified.
> - **Implementation**: Machine unlearning has been successfully applied in various real-world scenarios. As discussed in Section 2, it has also been utilized to mitigate copyright risks [9-10].
>
> We have included above discussion in Section 2 for better discussion.
>
> **References** (simplified due to word limitation)
>
> [1] Guo Wentao, et al. "Zeroth-order fine-tuning of llms with extreme sparsity"
>
> [2] Malladi Sadhika, et al. "Fine-tuning language models with just forward passes"
>
> [3] Zhao Jiawei, et al. "Galore: Memory-efficient llm training by gradient low-rank projection"
>
> [4] Azaria Amos, and Tom Mitchell. "The internal state of an LLM knows when it's lying"
>
> [5] Ji Ziwei, et al. "Llm internal states reveal hallucination risk faced with a query"
>
> [6] Chen Chao, et al. "INSIDE: LLMs' internal states retain the power of hallucination detection"
>
> [7] Chen Tong, et al. "CopyBench: Measuring literal and non-literal reproduction of copyright-protected text in language model generation"
>
> [8] Kurmanji Meghdad, et al. "Towards unbounded machine unlearning"
>
> [9] Dou Guangyao, et al. "Avoiding Copyright Infringement via Large Language Model Unlearning"
>
> [10] Yao Yuanshun, et al. "Large language model unlearning"

---

### Decision · Program_Chairs · 2025-04-30

**Decision:**

Accept (poster)

**Comment:**

Reviewers felt this was a very thorough exploration of managing copyright risks. The comprehensive discussion explores pre-deployment and post-deployment strategies, and discusses how they're each insufficient on their own. Weaknesses seemed relatively minor and substantial work to address.